# Planarians as an In Vivo Experimental Model for the Study of New Radioprotective Substances

**DOI:** 10.3390/antiox10111763

**Published:** 2021-11-04

**Authors:** Artem M. Ermakov, Kristina A. Kamenskikh, Olga N. Ermakova, Artem S. Blagodatsky, Anton L. Popov, Vladimir K. Ivanov

**Affiliations:** 1Institute of Theoretical and Experimental Biophysics, Russian Academy of Sciences, 142290 Pushchino, Russia; ermakovam@iteb.ru (A.M.E.); kamenskikhka@iteb.ru (K.A.K.); ermakovaon@iteb.ru (O.N.E.); blagodatskyas@iteb.ru (A.S.B.); popoval@iteb.ru (A.L.P.); 2Kurnakov Institute of General and Inorganic Chemistry, Russian Academy of Sciences, 119991 Moscow, Russia

**Keywords:** planarians, model animal, irradiation, regeneration, radioprotection

## Abstract

Ionising radiation causes the death of the most actively dividing cells, thus leading to depletion of the stem cell pool. Planarians are invertebrate flatworms that are unique in that their stem cells, called neoblasts, constantly replace old, damaged, or dying cells. Amenability to efficient RNAi treatments, the rapid development of clear phenotypes, and sensitivity to ionising radiation, combined with new genomic technologies, make planarians an outstanding tool for the discovery of potential radioprotective agents. In this work, using the well-known antioxidant N-acetylcysteine, planarians are, for the first time, shown to be an excellent model system for the fast and effective screening of novel radioprotective and radio-sensitising substances. In addition, a panel of measurable parameters that can be used for the study of radioprotective effects on this model is suggested.

## 1. Introduction

Radioprotectors are the substances that guard the body from damage to its molecules, organs, tissues, and cells in case of exposure to ionising radiation [1]. Their effect is based mainly on enhancing antioxidant cell defences by inactivating reactive oxygen species (ROS) and free radicals which emerge from water radiolysis [2]. Today, a large number of radioprotective substances are known that can reduce the negative effects of ionising radiation [3]. One of the most well-known antioxidant radioprotective agents is N-acetyl-L-cysteine (NAC), which belongs to the thiol compounds family [4]. It is commonly accepted that the effect of ionising radiation on biological objects leads to the formation of various forms of free radicals with different lifetimes [5]. Due to the presence of a large number of unpaired electrons, ROS have a high level of redox activity, which leads to oxidative damage in components of the cell [6]. The presence of a reduced thiol group in NAC can effectively neutralise free radicals and ROS generated by ionising radiation exposure [7,8,9]. There is also evidence that exogenous NAC can be a source of cysteine (Cys), which can be used for enhanced biosynthesis of intracellular glutathione (GSH), which also acts as a low molecular weight antioxidant [10]. Today, NAC is widely used as a reference compound in assessing the antioxidant properties of test substances in various oxidative stress models [11,12,13,14].

Despite significant progress in the development of radioprotective substances for military purposes, there is still a need for the development of radioprotectors and radiomitigators for medical applications, in particular for radiation therapy [15,16,17,18]. Using various approaches and molecular systems, it is also possible to enhance the effect of ionising radiation on tumour tissue by changing its radiosensitivity [19]. However, using common laboratory animals such as rats and mice, the search for such compounds is time-consuming and quite expensive. Thus, the development of new experimental models that are both relevant for biomedical purposes and capable of providing rapid screening with low cost is an urgent task. Planarians are model organisms that have a unique ability to regenerate due to the presence of neoblasts in their tissues [20]. Planarian neoblasts are totipotent stem cells that divide and differentiate into all types of cells in the adult body, including germline cells. The content of neoblasts in the planarian body reaches about 30% of the total cell number [21]. These cells endow the planarian body with an unlimited regenerative potential after damage or during organ self-renewal [22]. Moreover, planarian regeneration is possible, even from very small fragments, where at least one stem cell has been preserved [23]. The abovementioned features of planarians have made them a classic biological model for research into the regulation of stem cell proliferation and differentiation in vivo, restoration of differentiated tissue, and ageing [24]. In particular, the *Schmidtea mediterranea* planarian has a series of advantages over vertebrates as a model for biomedical research: the animals are cheap and easy to handle, they have a short life cycle, and they are available in large quantities.

It has previously been shown that neoblasts, like human stem cells, are sensitive to ionising radiation [25]. Irradiation of planarians in a dose of more than 15 Gy leads to the death of neoblasts, the inability to regenerate, and the termination of homeostatic tissue recovery [26]. The loss of neoblasts is accompanied by the characteristic abdominal curling of animals and further death within four weeks after irradiation [27]. Smaller doses of ionising radiation lead to the partial death of neoblasts, but the remaining part is able to restore the worm’s body and provide the ability for normal regeneration [28].

The effect of a sublethal irradiation dose on planarians, which preserves part of the neoblast and provides further regenerative potential, is the basis of the experimental model proposed here for studying potential radioprotective agents. Radioactivity is known to have pronounced and measurable effects on planarians, which can easily be tracked and quantified, such as on blastema regeneration rate, the number of surviving neoblasts and their transcriptional activity, the amount of DNA damage, and ROS generation. The current study has used a well-known antioxidant with proven radioprotective activity (N-acetylcysteine) in order to develop an experimental model of planarian radiosensitivity, select optimal irradiation doses and patterns, determine the timing of the effective assessment and establish a panel of trackable characteristics that can be used for analysing the radioprotective effect of a chemical substance on the planarian model.

## 2. Materials and Methods

### 2.1. Animals

The study featured an asexual laboratory strain of a freshwater flatworm *Schmidtea mediterranea* (*Turbellaria, Platyhelminthes*). The animals were kept at room temperature, in darkened glass aquariums containing a mixture of tap and distilled water at a 2:1 vol., which is the optimal ratio for keeping freshwater flatworms in lab conditions. Planarians were fed twice a week with mosquito larvae (*Chironomidae)*. Before the experiment, flatworms were starved for one week. This fasting stage is necessary to exclude the possible influence of nutritional components on the effect of X-ray radiation. This technique is generally accepted in planarian experiments [29].

For the experiments, the animals with nearly equal body lengths (about 8 mm) were selected. The anterior part of the planarian body (approximately 1/5 of the total length) containing the cephalic ganglion was cut off (decapitated) using a Carl Zeiss Stemi 2000 dissecting microscope, with a thin eye scalpel. Prior to decapitation, the planarians were placed on a cooling table for several minutes. The number of animals in each experimental group was the same (35 animals).

### 2.2. Computer-Assisted Morphometry In Vivo

The growth of regenerating blastema was studied using computer morphometry [30]. 72 h after decapitation, the images of control and experimental animals were taken using a Carl Zeiss Stemi 2000 microscope equipped with a Carl Zeiss AxioCam camera. To assess the blastema growth rate, the regeneration index R = s/S was used. The values of the blastema area (s) and total body area (S) were calculated using the Plana 4.0 software. 30 animals were used in each experimental or control group. Each experiment was repeated in triplicate. The relative change was calculated as follows:(1)$$\Delta R=\frac{\left(R_{E}-R_{c}\right)\pm \left(\delta_{E}-\delta_{c}\right)}{R_{c}}\times 100\%$$

Here, *R*_*E*_ is the index of regeneration (*R*) in the experimental group of flatworms; *R_C_* is the index of regeneration (*R*) in the control group of flatworms; Δ*R* is the difference (%) between *R**_E_* and *R**_C_*; *δ**_E_* and *δ**_C_* are standard errors of measurement in the experimental and control groups, respectively. The results presented here are mean values from three independent experiments.

### 2.3. Whole-Mount Immunocytochemical Study of Planarian Stem Cell Mitotic Activity

For this study, planarians with a body length of about 4 mm were selected. The number of mitotic cells in the regenerating worms was determined after seven days. Planarians were treated with 7% N-acetylcysteine solution for 5 min and fixed in PBS containing 4% formaldehyde and 0.3% Triton X100 for 20 min. Planarian staining for detecting mitotic cells was performed according to the protocol provided by Newmark and Alvarado [31]. To label mitotic cells, we used a primary antibody for phosphorylated histone H3 (Santa Cruz, Dallas, TX, USA), 1/1000 dilution. A secondary antibody conjugated to a fluorescent label CF488A (Biotium, Fremont, CA, USA) was used in 1/1000 dilution. Phosphorylated H3 histone has long been used as a classical marker of mitotic cells in studies of the planarian neoblast mitotic activity [32,33,34].

After washing in PBS, the whole-mount preparations were placed in Vectashield Antifade Mounting Medium (Vector Labs, Burlingame, CA, USA) and analysed using a Leica TCS SP5 confocal laser scanning microscope. The mitotic cell number and the planarian body area were measured using the Carl Zeiss Axio Image software. The number of mitotic cells per 1 mm^2^ of planarian body (the mitotic index) was then calculated. The results obtained were analysed statistically. The average values of the mitotic indices (i.e., the relationship of the total number of mitotic cells to the body area of each animal) were obtained using 10 animals per experimental group, in three experimental repetitions [35,36]. The specificity of immunocytochemical staining was confirmed using a non-immune serum. All controls were negative and demonstrated the absence of specific and non-specific fluorescent staining in planarian tissues.

### 2.4. Experimental Testing Substance

N-acetylcysteine (NAC) (Sigma, Burlington, MA, USA) at final concentrations of 10 and 15 mM was used as a test radioprotective agent. A stock solution of NAC (1 M, pH = 7) was prepared using distilled water. NAC was added to planarians 12 h before irradiation.

### 2.5. Planarian X-ray Irradiation

The planarians were irradiated using an X-ray machine, RUT-12 (15 mA, 200 kV). For irradiation, animals were placed in Petri dishes on filter paper moistened with water. The radiation doses were 1, 5, 10, 15, and 30 Gy at a power of 2 Gy per min.

### 2.6. RAPD PCR for Genotoxicity Analysis

Non-regenerating planarians were incubated with NAC, irradiated, and then genotoxicity analysis was performed using an approved protocol. A detailed description of the procedure was reported elsewhere [36]. For each primer, genomic template stability (GTS) was calculated:GTS (%) = (1 – *a*/*n*)·100%(2)
where *a* is the number of polymorphic bands detected in each treated sample and *n* is the number of total bands in the control. Polymorphism observed in a RAPD profile included the disappearance of a normal band or the appearance of a band in comparison with the control profile [37]. The sensitivity of the GTS parameter was calculated as a percentage of the control.

### 2.7. RT-PCR for Gene Expression Analysis

The level of elimination and restoration of the stem cell population was determined by changing the expression of 24 neoblast marker genes [38,39]. For this, mRNA was isolated from five planarians in the experimental and control groups by means of a set with magnetic particles, according to the attached protocol (Sileks, Moscow, Russia). Reverse transcription was performed with a Sileks (Russia) kit, using oligo dT primer, according to the attached protocol. The resulting cDNA served as a template for real-time PCR. The reaction was carried out using a reaction mixture with SybrGreen (Eurogen, Russia), on a CFX-96 thermocycler (BioRad, Philadelphia, PA, USA). The level of gene transcription was normalised by the average transcription levels of housekeeping genes *Smed-ef1* (GenBank accession number AY067688) and *Smed_01699* (GenBank accession number JX010505). Genomic DNA contamination was determined by the sample without the stage of reverse transcription based on genome-specific primers. Gene-specific primers were selected using the Primer Express program (Applied Biosystems, Waltham, MA, USA).

The expression data were analysed using the online service http://www.qiagen.com (accessed on 25 September 2020), the mayday-2.14 program (Center for Bioinformatics, Tübingen, Germany), and the Genesis program [40]. Only those results were taken into account for which changes in the level of gene expression were observed at *p* < 0.05.

### 2.8. ROS Measurement in the Planarian Body

ROS levels in the planarian body after irradiation were identified using H2DCFDA (2,7-dichloro-dihydrofluorescein-diacetate-acetyl). This dye is a well-known fluorescent intracellular sensor of active oxygen species [41]. Animals were placed in a solution of 10 μM -H_2_DCFDA (Biotium, USA) and incubated for 60 min in the dark. Next, the planarians were incubated with NAC, washed twice with water, and irradiated using an X-ray machine. The positive control group was obtained by pre-incubation of animals for 30 min in 100 μM H_2_O_2_ (Sigma, USA). Then, the planarians were anesthetised for 5–10 min in a 0.1% solution of chloroethane (Sigma, USA) [42] and photographed with an Axio Scope A1 fluorescence microscope (Carl Zeiss) (Ex/Em = 492–495/517–527 nm). In the images obtained using the ImageJ program (National Institute of Health, Bethesda, MD, USA), the total fluorescence intensity of the animal body was estimated. The measurement results were averaged over 15 animals.

### 2.9. Statistical Data Analysis

The data obtained were treated statistically by the Sigma-Plot 9.11 program (Systat Software Inc., Erkrath, Germany) using one–way ANOVA analyses of variance.

### 2.10. Ethical Standards

All procedures performed in this study involving animals were performed in accordance with the ethical standards of the institution at which the studies were conducted.

## 3. Results and Discussion

### 3.1. Working Dose Selection

Firstly, to select the dose range for further experiments, irradiation was carried out on decapitated regenerating planarians with a series of gradually increasing doses and the regeneration rate, mitotic activity, and DNA damage were monitored. Dose-dependent effects on the growth process of the planarian head blastema under X-ray irradiation were found (Figure 1). The inhibition of blastema growth was observed after 5, 10, 15, and 30 Gy irradiation doses. The blastema size after 5 Gy irradiation was 32% less than in the unirradiated control group. An increase of irradiation dose to 10, 15, and 30 Gy led to the inhibition of blastema regeneration in a dose-dependent manner by 45, 63, and 83%, respectively.

An assessment of stem cell mitotic activity on the third day after irradiation revealed a complete absence of mitotic cells after 15 and 30 Gy irradiation. At lower irradiation doses (1, 5, or 10 Gy), the mitotic activity of the neoblasts was still present in the planarian body.

To evaluate the DNA damage caused by irradiation, the randomly amplified polymorphic DNA (RAPD)-PCR technique was used. RAPD is a PCR-based method that amplifies random DNA fragments through the use of single, short primers of arbitrary nucleotide sequence, under low annealing conditions. This method is widely used to determine genetic polymorphism in populations and DNA damage from genotoxic chemical or physical factors [37]. The genomic template stability (GTS) parameter reflects the difference between the control and the treated samples: a larger GTS value means greater similarity, i.e., less damage caused to the DNA by irradiation [37].

After 10 and 15 Gy irradiation, significant changes were observed in genomic template stability (GTS). In particular, GTS after 10 Gy irradiation dropped down to 71%, whereas after irradiation at a dose of 15 Gy, it made up 60%.

On the basis of the abovementioned observations, the working doses were selected for experiments on radioprotection. A dose of 5 Gy irradiation did not cause notable changes in the blastema growth rate, but a 30 Gy irradiation dose led to a significant slowdown in regeneration, up to complete inhibition. Only medium irradiation doses (10 and 15 Gy) yielded a significant slowdown in head regeneration, a decrease in the mitotic activity of stem cells, and damage in the DNA of treated planarians. Based on the results, only two irradiation doses were selected—10 and 15 Gy—for further studies of radioprotective action.

### 3.2. Blastema Growth Rate in Irradiated Planarians Was Increased by NAC

Next, NAC was introduced to test its radioprotective properties on regenerating planarians after irradiation. The first and most straightforward parameter to control was regenerating blastema growth rate. Pre-incubation of planarians in NAC solution in two different concentrations (10 and 15 mM) before X-ray irradiation led to an increase in blastema growth rate by the third day of regeneration (Figure 2). Statistically significant effects of NAC radioprotection were observed on days 3–7. Thus, NAC significantly improved the dynamics of regeneration and had a radioprotective effect on the planarian model.

### 3.3. Neoblast Survival and Mitotic Activity Was Increased by NAC

To reveal the physiological mechanisms of NAC protective action, an analysis was made of the activity of the neoblasts after exposure to X-ray radiation (Figure 3). It was found that the recovery of regeneration potential directly correlated with the number of neoblasts in the planarian body after irradiation. An analysis of mitotic activity showed that there were no mitotic cells in the planarian body after irradiation at a dose of 15 Gy on day 7. When using NAC as a radioprotector (10 mM), the planarians retained about 10% of mitotically active neoblasts (Figure 3a,b). On the tenth day after irradiation (15 Gy), mitotic cells were observed mainly in the head while with NAC as a radioprotector they were found not only in the head, but also in the pharyngeal and caudal parts of the body (Figure 3c). It is known that doses up to 15 Gy are sublethal for planarians, and doses above 20–30 Gy are lethal. At lethal doses, the few surviving neoblasts completely lose their proliferation ability [43,44]. Note, radiation-induced death of planarian stem cells is probably due to the same mechanisms (DNA damage, repair, apoptosis) that have been described for mammalian stem cells [45,46]. At sublethal doses of radiation which were used in our study, the surviving neoblasts were still able to give rise to new clonal populations [47], but this process is quite slow. Therefore, when the decapitation is done immediately after irradiation, the deficiency of stem cells significantly reduces the regeneration rate of planaria and the blastema growth rate. The presence of NAC radioprotector in the planaria irradiated with sublethal doses preserves higher content of neoblasts which are able for further proliferation.

### 3.4. NAC Increased the Neoblast Markers Expression and Reduced DNA Damage

Gene expression analysis of planarian neoblast markers showed that, on the third and sixth day after X-ray irradiation, the concentration of the studied mRNAs significantly decreased in treated animals (Figure 4). The expression of the *Smed-soxP-1*, *Smed-fgfr-4*, *Smed-gata456,* and *Smed-hnf-4* genes was higher with NAC after a 15 Gy irradiation dose when compared to the control group [39].

Similar results were obtained for the stability of planarian genomic DNA after X-ray irradiation. The damaging effect of radiation on the genomic DNA was least pronounced after treatment of the animals with NAC, which reduced the degree of change and increased genomic stability (Figure 5).

### 3.5. The Amount of ROS Generated after Irradiation Was Reduced by NAC

Another parameter critical to control after irradiation is reactive oxygen species (ROS) generation. Results of ROS measurement in a planarian body after X-ray irradiation are shown in Figure 6. X-ray irradiation led to the formation of free radicals in a dose-dependent manner, therefore the minimum autofluorescence intensity of the planarian body was observed in the control group. Irradiation led to a significant increase in the amount of ROS, which led to an increase in fluorescence intensity. It is also worth noting that neoblasts are mainly located in the parenchyma zone, which actively fluoresces after irradiation. Such colocalization of high ROS levels and neoblasts in the planarian body after irradiation confirms oxidative damage to stem cells, which affects their proliferation and migration. Pre-incubation of planarians with N-acetylcysteine (10 mM) resulted in a significant decrease in the level of dye fluorescence after irradiation of animals in various doses (5, 10, and 15 Gy). These results confirm directly that NAC acts as an antioxidant, effectively inactivating ROS under X-ray irradiation.

## 4. Conclusions

In this study, a series of experiments was conducted proving that planarians provide a simple and suitable model for studying radioactive damage and the radioprotective effects of chemical substances. Using a well-known radioprotector, N-acetylcysteine, as a model substance, it has been shown that planarians possess a set of easily measurable characteristics which can be used to quantify radiation damage and the radioprotective effect. These are: (i) the growth rate of regenerating blastema, (ii) the number of surviving neoblasts and their mitotic activity, (iii) the expression of neoblast marker genes, (iv) the amount of DNA damage, and (v) the rate of ROS generation by the planarian tissue.

The first parameter can be used to directly evaluate the efficacy of radioprotection. This criterion is unique for planarians as model animals; it is simple but requires a microscope and software. The next two parameters investigate stem cell biology in greater depth and reflect the stage of neoblasts—the cells which are responsible for planarian regeneration. Monitoring the status of neoblasts is important, for radiation mostly damages fast dividing cells and neoblasts can serve as a perfect model thereof. The fourth parameter enables the measurement of the impact of radiation and the efficacy of radioprotectors on genotoxicity. Finally, the last of these parameters assess the molecular mechanisms of ionising radiation damage. Taken together, this set of simple and robust parameters makes it possible to screen and characterise potential radioprotectors in an inexpensive and robust manner. There are many new potential radioprotective substances among natural products [48], therefore the screening system described here has great potential. The search for radiosensitisers is also a task of high importance, for they enable radiation harm to cancer cells to be increased without affecting normal cells [49]. The model system described in the current study is also applicable for the screening of radiosensitisers.

In general, the observed effects of radiation-induced suppression of planarian regeneration are associated with partial or complete elimination of the neoblast population after X-ray radiation exposure. The results reported here show that the remaining pool of neoblasts after irradiation gives rise to a new population of stem cells and thus ensures the regeneration of the planarian body. The molecular mechanisms of neoblast proliferation, migration, and differentiation have been extensively studied using modern methods, including RNA interference (RNAi) for gene-specific knockdown [50]. This makes it possible to clearly monitor the influence of external factors and stimuli, including ionising radiation or a model drug action, on the planarian’s vital processes. The radioprotective effect of N-acetylcysteine is based on the suppression of ROS formation during irradiation, which enables the saving of a fairly large number of neoblasts in the body of the planarian. Obviously, the survival of stem cells when using a radioprotector is associated not only with direct antioxidant protection, but also with the effect on the DNA repair rate. The results have demonstrated the possibility of using planarians as a convenient model for studying the radioprotective properties of various substances.

## Figures and Tables

**Figure 1 antioxidants-10-01763-f001:**
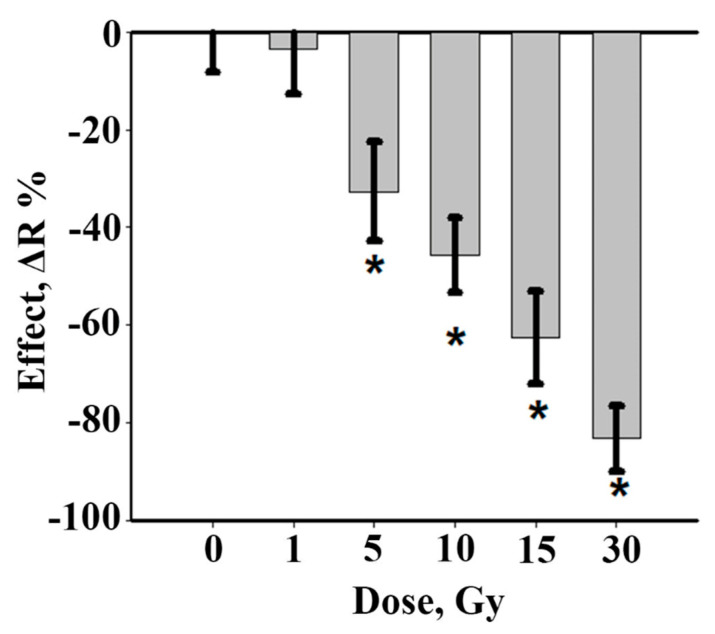
Inhibition of planarian head blastema growth on the 3rd day after X-ray irradiation (1—30 Gy). Data are shown as mean values ± standard error, *n* = 90, * *p* < 0.001.

**Figure 2 antioxidants-10-01763-f002:**
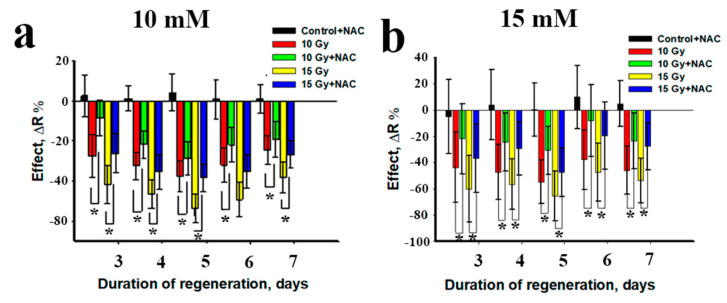
The radioprotective effect of NAC under X-ray exposure. (**a**) Radioprotective effect of N-acetylcysteine (10 mM) under 10 and 15 Gy irradiation; (**b**) radioprotective effect of N-acetylcysteine (15 mM) under 10 and 15 Gy irradiation, * *p* < 0.001 via ANOVA.

**Figure 3 antioxidants-10-01763-f003:**
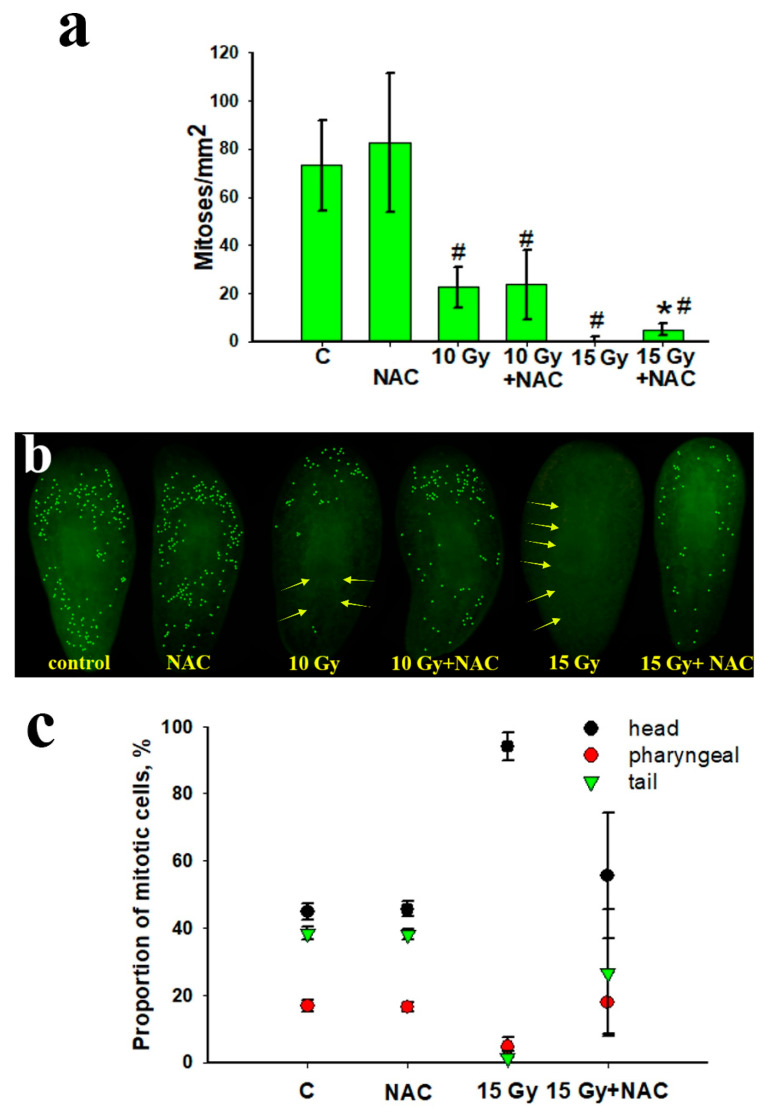
Number of mitotic cells in the planarian body 7 days after X-ray irradiation (10 and 15 Gy) with NAC (10 mM). (**a**,**b**) Determination of the total number of mitotic cells in the planarian body stained by immunohistochemistry; (**c**) distribution of mitotic cells in the planarian body 10 days after irradiation at a dose of 15 Gy. # *p* < 0.001 (from control), * *p* < 0.001 (from 15 Gy). Arrows indicate the absence of mitotic cells after X-ray irradiation.

**Figure 4 antioxidants-10-01763-f004:**
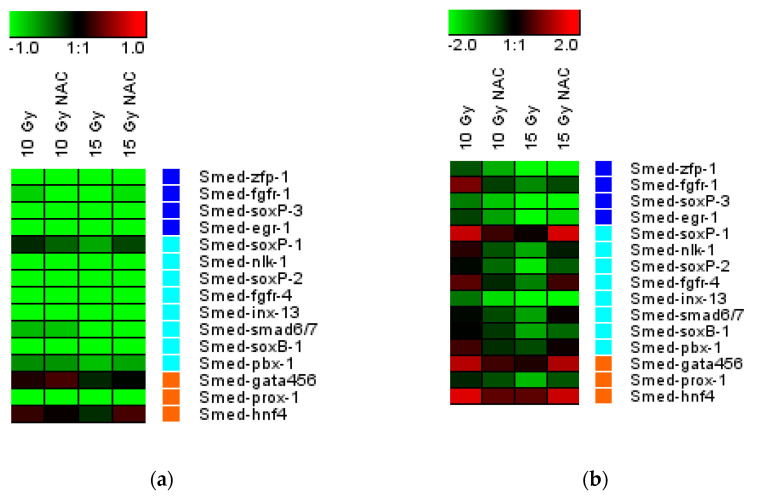
Gene expression of three classes of neoblast markers in regenerating planarians treated with NAC (10 mM) for 3 h (**a**) and 6 h (**b**) after irradiation. The intensity scale of the standardised expression values range from −3 (green: low expression) to +3 (red: high expression), with a 1:1 intensity value (black) representing the control (unirradiated).

**Figure 5 antioxidants-10-01763-f005:**
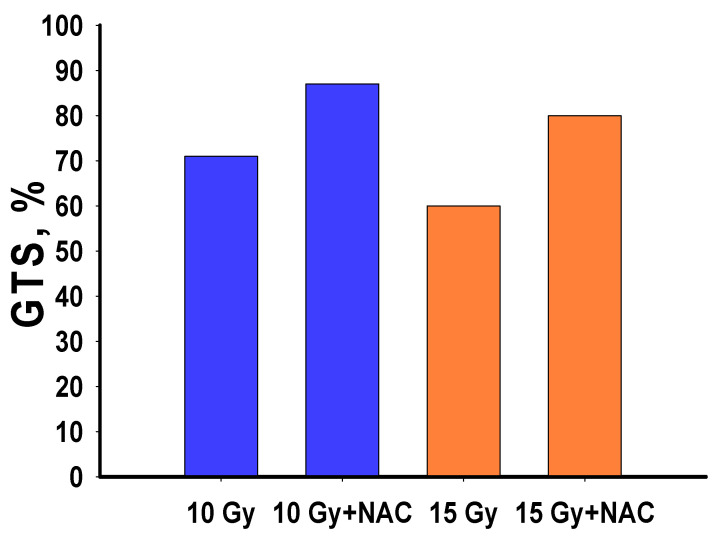
Genomic template stability (GTS) of planarians after X-ray irradiation (DNA was isolated 2 h after irradiation). GTS—genomic stability coefficient.

**Figure 6 antioxidants-10-01763-f006:**
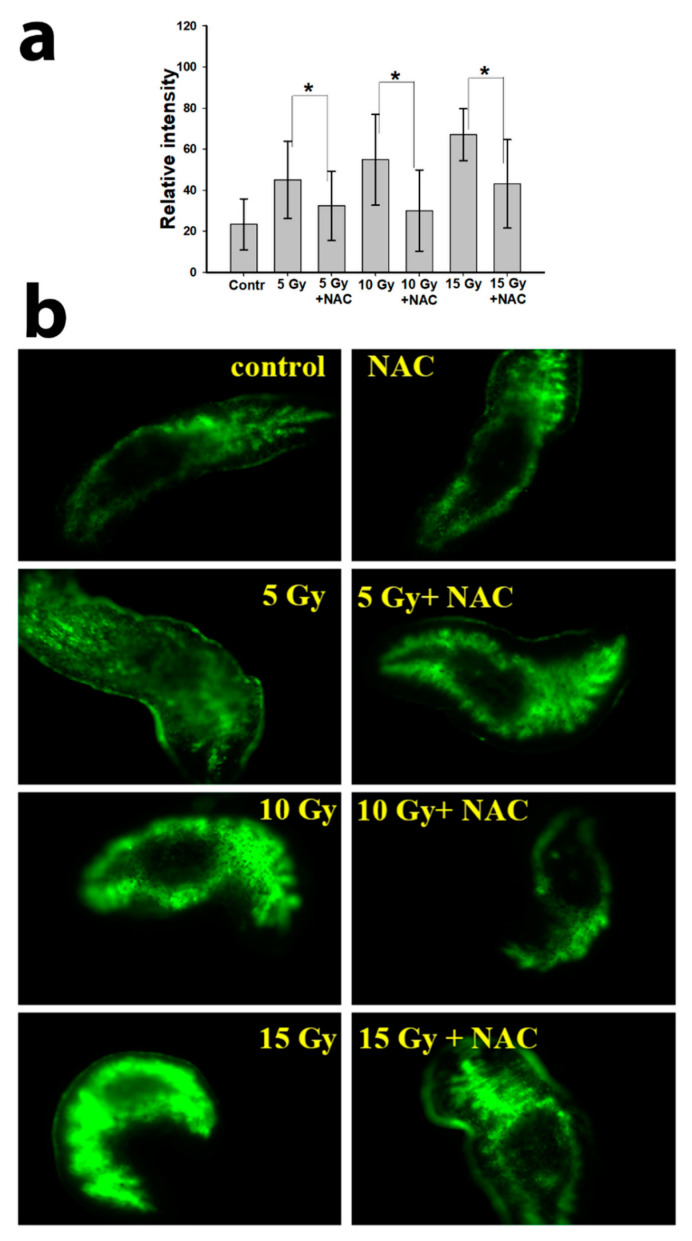
Inhibition of ROS formation by N-acetylcysteine, measured by H2DCFDA fluorescence. Quantitative determination of the fluorescence intensity in the body of planarians (**a**); fluorescence micrographs of planarians after irradiation (**b**). Standard deviation * *p* < 0.001.

## Data Availability

The data presented in this study are available in article.

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
