# Peer review of "Planarians as an In Vivo Experimental Model for the Study of New Radioprotective Substances"

_antioxidants, 2021, doi:10.3390/antiox10111763_

Round 1

Reviewer 1 Report

The presented manuscript describes approaches to measure/quantify the effects of radioprotective substances on Planarians. Each approach sounds reasonable, and altogether it appears convincing.

It is an interesting and promising step to analyze radiation damage on whole organisms as they have done it.

phosphorylated H3 histone

please insert a reference why this is a marker for mitosis and not "only" for transcriptionally active loci

Figure 2

maybe for better understanding what are the changes int he setups for a and b, you might write "10mM" and "15mM" also in headlines of the graphs

Figure 3b is not well "readable"...maybe add some arrows to show what is supposed being interesting for the reader

maybe you might expand description and interpretation on Figure 3 a bit

Figure 5

please choose between "stability" and "instability" consistently when describing "GTS"

Figure 6

in b you have signals from all over the body...is there a pattern? what is interesting in these body scans? how does this corralate with findings decribed in Figure 3b? 

Author Response

We are very grateful for the reviewers’ comments, aimed at improving our paper. We carefully checked all the points and we tried to address all the questions and suggestions. Please find our comments below.

Referee 1

General comments: The presented manuscript describes approaches to measure/quantify the effects of radioprotective substances on Planarians. Each approach sounds reasonable, and altogether it appears convincing. It is an interesting and promising step to analyze radiation damage on whole organisms as they have done it.

Discussion: We are very grateful for the reviewer for the positive evaluation of our work.

Issue 1: phosphorylated H3 histone. Please insert a reference why this is a marker for mitosis and not "only" for transcriptionally active loci

Discussion: We thank the reviewer for the very valuable comment.

Corrections made in the manuscript: In the Materials and Methods section of the manuscript, we included the following text as well as the relevant references: “Phosphorylated H3 histone is a commonly used marker of the mitotic activity of the planarian neoblasts [32-34].

Issue 2:  Figure 2. maybe for better understanding what are the changes in the setups for a and b, you might write "10mM" and "15mM" also in headlines of the graphs

Discussion: We thank the reviewer for the valuable comment.

Corrections made in the manuscript: In Figure 2, we added the concentration values (10 mM and 15 mM).

Issue 3: Figure 3b is not well "readable"...maybe add some arrows to show what is supposed being interesting for the reader

Discussion: We thank the reviewer for the very valuable comment.

Corrections made in the manuscript: We improved the quality of the Figure, and marked the areas without neoblasts with arrows. We also added the text in the figure caption: “Areas without neoblasts are marked with arrows”.

Issue 4: maybe you might expand description and interpretation on Figure 3 a bit

Discussion: We thank the reviewer for the very valuable comment.

Corrections made in the manuscript: In the Results section of the manuscript, we included the following text: “It is known that doses up to 15 Gy are sublethal for planarians, and doses above 20-30 Gy are lethal. At lethal doses, the few survived neoblasts completely lose proliferation ability [38, 39]. Note, radiation-induced death of planarian stem cells is probably due to the same mechanisms (DNA damage, repair, apoptosis) that have been described for mammalian stem cells [41, 42]. At sublethal doses of radiation which were used in our study, the survived neoblasts are still able to give rise to new clonal populations [40], but this process is quite slow. Therefore, when the decapitation is done immediately after irradiation, the deficiency of stem cells reduces significantly the regeneration rate of planaria and the blastema growth rate. The presence of NAC radioprotector in the planaria irradiated with sublethal doses preserves higher content of neoblasts which are able for further proliferation.

Issue 5: Figure 5. Please choose between "stability" and "instability" consistently when describing "GTS"

Discussion: We thank the reviewer for drawing our attention to this misprint.

Corrections made in the manuscript: In the figure caption, we corrected “instability” for “stability”.

Issue 6: Figure 6: in b you have signals from all over the body...is there a pattern? what is interesting in these body scans? how does this correlate with findings decribed in Figure 3b? 

Discussion: We thank the reviewer for the very valuable comment. Planaria was exposed to X-ray irradiation, therefore, the H2DCFDA dye indicates the generation of ROS throughout their bodies. A dose-dependent increase in the fluorescence intensity of the ROS indicator dye is observed, the quantitative assessment of this parameter is presented in Figure 6a. Note, the neoblasts are mainly located in the parenchyma zone, which actively fluoresces after irradiation. Such colocalization of high ROS levels and neoblasts in the planarian body after irradiation confirms oxidative damage to stem cells, which affects their proliferation and migration.

Corrections made in the manuscript: In the Results section of the manuscript, we included the following text: “It is also worth noting that the neoblasts are mainly located in the parenchyma zone, which actively fluoresces after X-ray irradiation. Such colocalization of high ROS levels and neoblasts in the planarian body after irradiation confirms oxidative damage to stem cells, which affects their proliferation and migration.”

Reviewer 2 Report

The authors of the paper propose screening of radioprotective substances in the experiments on planarians and present the results of the first experiment of such a kind with N-acetylcysteine. They also mention the possibility of similar screening of radiosensitisers. The proposal is interesting and useful for radiobiology and medicine. The paper can be published in Antioxidants after minor editing. The necessity or desirability of editing of several sentences results from the following:

  1. The word “reliable” in the Abstract (page 1, line 18) should be either removed or briefly explained.
  2. It seems to be useful to mention screening of radiosensitisers in Abstract.
  3. It is desirable to present at least one reference on paper or book in the sentence in which radioprotective substances used for military purposes are mentioned (page 1, lines 41-43).
  4. The situation with bioethical restrictions is not clear (see page 2, line 61, and page 5, line 199).
  5. It is desirable to explain why “Before the experiments, flatworms were starved for one week” (page 2, lines 86-87).
  6. The word “about” should probably be inserted before “4 mm” on page 3, line 111 (see also page 2, line 87).
  7. List of authors contains 6 names, while Section “Authors contribution”(page 11) describes contributions of 7 authors. Besides, it seems that “V.K.I.” should be written instead of “I.V.K.”.

Author Response

We are very grateful for the reviewers’ comments, aimed at improving our paper. We carefully checked all the points and we tried to address all the questions and suggestions. Please find our comments below.

Referee 2

General comments: The authors of the paper propose screening of radioprotective substances in the experiments on planarians and present the results of the first experiment of such a kind with N-acetylcysteine. They also mention the possibility of similar screening of radiosensitisers. The proposal is interesting and useful for radiobiology and medicine. The paper can be published in Antioxidants after minor editing. The necessity or desirability of editing of several sentences results from the following:

 Discussion: We are very grateful for the reviewer for the positive evaluation of our work.

Issue 1: The word “reliable” in the Abstract (page 1, line 18) should be either removed or briefly explained.

Discussion: We thank the reviewer for the comment.

Corrections made in the manuscript: We removed this word.

Issue 2: It seems to be useful to mention screening of radiosensitisers in Abstract.

Discussion: We thank the reviewer for the very valuable comment.

Corrections made in the manuscript: We radiosensitizers in the Abstract.

Issue 3: It is desirable to present at least one reference on paper or book in the sentence in which radioprotective substances used for military purposes are mentioned (page 1, lines 41-43).

Discussion: We thank the reviewer for the very valuable comment.

Corrections made in the manuscript: We added four relevant references:

  1. Kouvaris J. R., Kouloulias V. E., Vlahos L. J. Amifostine: the first selective-target and broad-spectrum radioprotector Oncologist 2007;12(6):738-47.
  2. Singh V. K., Seed T. M. The efficacy and safety of amifostine for the acute radiation syndrome Expert opinion on drug safety 2019;18(11):1077-1090
  3. Du J., Zhang P., Cheng Y., Liu R., Liu H., Gao F., Shi C., Liu C. General principles of developing novel radioprotective agents for nuclear emergency Radiation Medicine and Protection 2020;1(3):120–126
  4. Rosen E.M., Day R., Singh V. K. New approaches to radiation protection Oncol. 2015; 4, Article 381 

Issue 4: The situation with bioethical restrictions is not clear (see page 2, line 61, and page 5, line 199).

Discussion: We thank the reviewer for the very valuable comment.

Corrections made in the manuscript: From the introduction section, we removed the text: “and they are not subject to bioethical restrictions”. Experimental work with planarian flatworms does not require official approval from the bioethical committee, such approvals are applicable only for the experiments with mammals.

Issue 5: It is desirable to explain why “Before the experiments, flatworms were starved for one week” (page 2, lines 86-87).

Discussion: We thank the reviewer for the very valuable comment.

Corrections made in the manuscript: In the Materials and methods section of the manuscript, we included the following text: “This fasting stage is necessary to exclude the possible influence of nutritional components on the effect of X-ray radiation. This technique is generally accepted in experiments with planaria [29].

Issue 6: The word “about” should probably be inserted before “4 mm” on page 3, line 111 (see also page 2, line 87).

Discussion: We thank the reviewer for the comment.

Corrections made in the manuscript: We added the word “about”.

Issue 7: List of authors contains 6 names, while Section “Authors contribution”(page 11) describes contributions of 7 authors. Besides, it seems that “V.K.I.” should be written instead of “I.V.K.”.

Discussion: We thank the reviewer for the very valuable comment.

Corrections made in the manuscript: We changed the number of authors and corrected the abbreviation from (I.V.K.) to (V.K.I.).
